# Novel Techniques for Void Filling in Glacier Elevation Change Data Sets

**Thorsten Seehaus** [1],*, **Veniamin I. Morgenshtern** [2], **Fabian Hübner** [2], **Eberhard Bänsch** [3] **and Matthias H. Braun** [1]

1   Institute of Geography, Friedrich-Alexander University Erlangen-Nürnberg, 91054 Erlangen, Germany; matthias.h.braun@fau.de
2   Chair of Multimedia Communications and Signal Processing, Friedrich-Alexander University Erlangen-Nürnberg, 91054 Erlangen, Germany; veniamin.morgenshtern@fau.de (V.I.M.); huebner.fa@gmx.de (F.H.)
3   Department of Mathematics, Friedrich-Alexander University Erlangen-Nürnberg, 91054 Erlangen, Germany; baensch@math.fau.de
*   Correspondence: thorsten.seehaus@fau.de

**Abstract:** The increasing availability of digital elevation models (DEMs) facilitates the monitoring of glacier mass balances on local and regional scales. Geodetic glacier mass balances are obtained by differentiating DEMs. However, these computations are usually affected by voids in the derived elevation change data sets. Different approaches, using spatial statistics or interpolation techniques, were developed to account for these voids in glacier mass balance estimations. In this study, we apply novel void filling techniques, which are typically used for the reconstruction and retouche of images and photos, for the first time on elevation change maps. We selected 6210 km$^2$ of glacier area in southeast Alaska, USA, covered by two void-free DEMs as the study site to test different inpainting methods. Different artificially voided setups were generated using manually defined voids and a correlation mask based on stereoscopic processing of Advanced Spaceborne Thermal Emission and Reflection Radiometer (ASTER) acquisition. Three "novel" (Telea, Navier–Stokes and shearlet) as well as three "classical" (bilinear interpolation, local and global hypsometric methods) void filling approaches for glacier elevation data sets were implemented and evaluated. The hypsometric approaches showed, in general, the worst performance, leading to high average and local offsets. Telea and Navier–Stokes void filling showed an overall stable and reasonable quality. The best results are obtained for shearlet and bilinear void filling, if certain criteria are met. Considering also computational costs and feasibility, we recommend using the bilinear void filling method in glacier volume change analyses. Moreover, we propose and validate a formula to estimate the uncertainties caused by void filling in glacier volume change computations. The formula is transferable to other study sites, where no ground truth data on the void areas exist, and leads to higher accuracy of the error estimates on void-filled areas. In the spirit of reproducible research, we publish a software repository with the implementation of the novel void filling algorithms and the code reproducing the statistical analysis of the data, along with the data sets themselves.

**Keywords:** glacier mass balance; elevation change; void filling

## 1. Introduction

Imagery from remote sensing missions provides the ideal data set to carry out regional- to continental-scale analysis of glacier changes. Typical quantitative products derived from imaging sensors are glacier surface velocities (e.g., [1]) and surface elevation change maps (e.g., [2]). These data

sets are usually affected by voids, causing limitations for the direct assimilation into glacier models or for the computations of ice volume and mass changes.

Glacier mass balance is an important parameter for glacier change analysis and for the quantification of sources of global sea level change. Geodetic glacier mass balance is derived by differencing digital elevation models (DEMs) from different dates, integrating the measured elevation changes throughout the glacier areas and applying a volume to mass conversion factor [3]. Depending on the sensor, processing technique and study region, the coverage of glaciers by valid elevation change measurements derived from DEMs can vary between 43% and 97% for large-scale analyses (e.g., [4–6]). Therefore, different void filling approaches were developed to facilitate glacier-wide mass balance computations.

Data voids in products derived from optical remote sensing imagery are often caused by sensor saturation and low pixel correlation (both often occurring in the accumulation zones of glaciers) (e.g., [7]). Additionally, natural limitations like shadows and, in particular, cloud cover limit the data coverage (e.g., [5]). Synthetic-aperture radar (SAR) acquisitions have only minimal constraints due to meteorological conditions. However, the nature of the SAR imaging geometry (side-looking acquisitions) leads to data voids especially in mountainous regions, due to SAR layover and shadowing (e.g., [8]).

In this study, we focus on void filling in glaciological surface elevation change products based on DEMs derived from remote sensing acquisitions. DEMs or remote sensing data to generate DEMs with sufficient quality and spatio-temporal coverage have become increasingly more available in recent decades, allowing the analysis of glacier evolution on large temporal and spatial scales.

The study by McNabb et al. [9] provides a detailed review and evaluation of various void filling methods used in the literature for elevation change maps; we briefly describe these methods next.

Spatial methods either use bilinear interpolation of the voids in either the individual DEMs or the elevation change maps, or use average elevation differences of a certain region around the voids (e.g., [10,11]). Constant methods use the average elevation change on the measured glacier area multiplied by the total glacier area [12]. Hypsometric methods, mostly applied by recent regional analyses, determine the altitude dependence of the elevation changes and use this information to fill the voids in the respective altitudes (e.g., [2,4,5]). In the following, the abovementioned approaches are called *"classical"* void filling techniques.

In this study, we aim to evaluate algorithms for the void filling of glacier surface elevation data sets, which were already successfully applied for the reconstruction and retouche of images and photos. These techniques try to seamlessly fill the voids in an image, so that the interpolation is not identifiable by a viewer [13]. In the following, the approaches based on this idea are called *"novel"* void filling techniques. Several approaches have been developed and can be grouped into four main categories: geometric synthesis, texture synthesis, compressed sensing-based, and neural network-based methods. Geometric methods are either based on an iterative solution of, e.g., a partial differential equation or use coherence in the original image (e.g., [14]). Texture synthesis methods are based on the self-similarity principle. For example, they search the image for appropriate patches best suited to derive a synthetic texture to fill a void [15,16]. Prominent applications of this strategy are the Photoshop "Patchmatch" and "Content Aware Fill" tools or the GIMP "resynthesizer lib". There are also methods that analyze the whole image (or subsets of large images) to synthesize information in the voids [17]. The compressed sensing-based approaches are a further development of the latter approach, trying to reconstruct the void areas by extrapolating the information from around the void and using the fact that images are highly compressible when represented in an appropriate way (i.e., the number of coefficients needed to describe an image in, for example, wavelet or shearlet domains is much smaller than the number of pixels in that image). The neural network-based methods (e.g., [18]) represent the void filling function in a hierarchical nonlinear form parameterized by millions of parameters; these parameters are then determined based on an appropriate training data set. As explained above, certain glacier regions, such as regions that do not have surface texture, will have voids in all available data sets, limiting the

availability of adequate training data for such regions. Further, the algorithms are not designed for solving the void filling problems of glacier surface elevation change maps and are hard to adapt.

We selected three novel techniques using geometric synthesis and compressed sensing-based strategies as well as three classical void filling approaches, recommended by [9], and applied them to an artificially voided surface elevation change data set on a glacier region in southeast Alaska, which was also used by [9] (see Section 3.1). Our aim is to evaluate the reconstruction of the original information by the void filling algorithms and the impact on subsequent glacier volume change computations or model assimilation. Additionally, we perform a comparison of the results of the novel void filling techniques with those of recommended classical void filling techniques [9] in order to assess the achievable improvement.

## 2. Study Site and Data Set

A suitable void-free data set is needed to carry out an evaluation of void filling techniques, which may be conducted by introducing artificial voids into the original data and then using an algorithm to fill them. A recent analysis [9] tested classical void filling approaches using a large-scale void-free glacier surface elevation change data set from southeast Alaska, USA. Their study area includes a diverse sample of glacier geometries and glacier elevation change patterns. We carry out our analysis using the same study region and glacier surface elevation change data set as [9], facilitating the comparison of the novel and classical void filling approaches.

The study site is located at Glacier Bay and Lynn Canal, Alaska, USA (see Figure 1). It covers a glacier area of ~6210 km$^2$ consisting of over 700 individual glacier catchments according to the Randolph Glacier Inventory (RGI) 6.0 [19]. The ice-covered areas range from sea level to more than 4000 m a.s.l. Various glacier types such as surge-type glaciers and large and small valley glaciers as well as retreating and advancing tidewater glaciers can be found within the analyzed area.

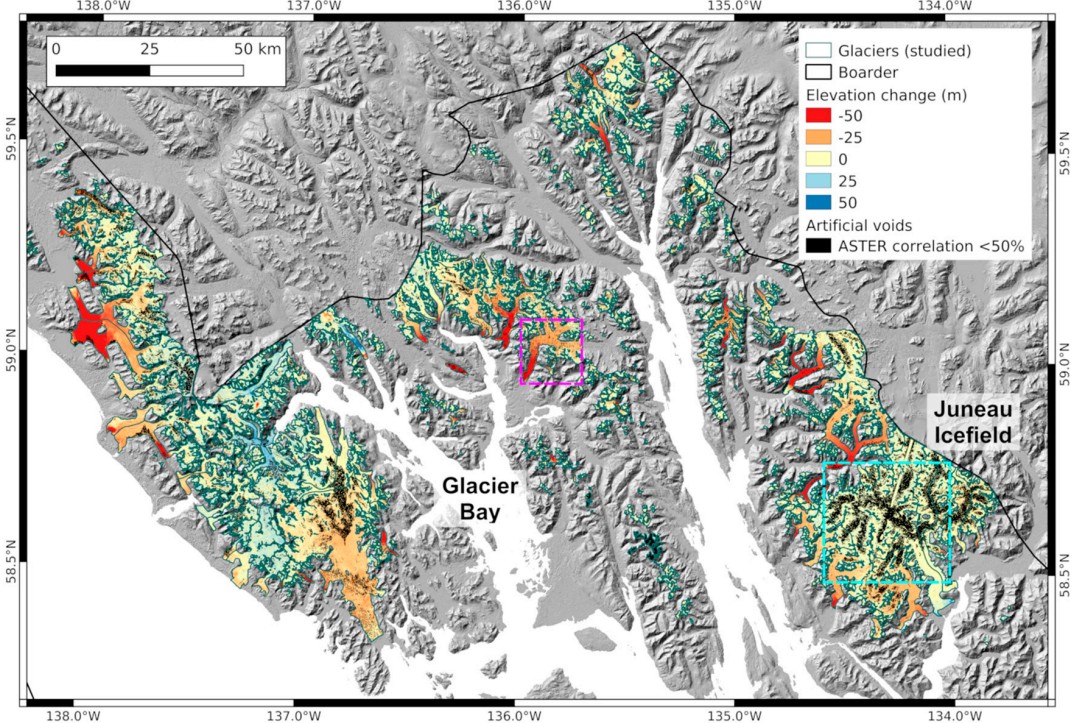

**Figure 1.** Overview of study area in southeast Alaska, USA. Glacier outlines are based on the Randolph Glacier Inventory 6.0. Elevation change is derived from SRTM (2000) and IfSAR (2012/13) data. Black areas indicate artificial voids on glacier surfaces using an ASTER correlation threshold of 50%. Background: ASTER GDEMV2 hillshade. Light blue dashed rectangle indicates the extent of the Juneau setup and pink dashed rectangle indicates the extent of the Center setup (see Figure 2).

The Shuttle Radar Topography Mission (SRTM) C-band global 1 arcsec DEM is used as elevation reference. It has a nearly complete coverage of the landmasses between 60° N and 54° S and was acquired within an eleven-day period in February 2000. Many geodetic mass balance studies rely on this data set (e.g., [4,11,20]). Radar layover and shadowing effects, due to the acquisition geometry of SRTM and the topography of the study site, caused voids in the SRTM DEM. Less than 2.5% of the glacier area is affected by voids in the SRTM data set. These regions with voids are not considered in our analysis, but need to be considered if one aims to carry out regional mass balance analysis, which is beyond the scope of this study.

Within the framework of the Statewide Digital Mapping Initiative, an interferometric synthetic-aperture radar (IfSAR) DEM throughout Alaska is generated by the state of Alaska using X and P band acquisitions. The study site was covered by survey flights in August 2012 and 2013. Glacier delineations are based on the RGI 6.0 and were manually refined by late summer Landsat imagery in 1999 and 2001, to obtain outlines close to the SRTM data (see [9] for a more detailed description of the data sets).

## 3. Methods

### 3.1. Void-Free Volume Change and Artificial Voids

Voids in elevation change fields can be caused by missing data in each of the DEMs used to compute the changes. One has the option to fill in the voids in the individual DEMs or directly in the elevation change fields. We chose the latter option and decided to test and evaluate the void filling on the glacier elevation change fields for the following reasons. Typically on mountain glaciers, the magnitude and the gradient of elevation change fields are much smaller than the magnitude and the gradient of the elevation fields themselves. Thus, a less variable pattern needs to be reconstructed by the void filling approaches applied to elevation change fields. Moreover, the void filling needs to be performed only for one data set, making the analysis more intuitive and less computationally expensive.

The void-free glacier surface elevation change field with 30 m resolution was kindly provided by McNabb et al. [9]. More information on the processing can be found in the respective publication. By integrating the elevation change information over the glacier areas, the volume changes for individual glaciers and for the whole study area are computed. Similar to McNabb et al. [9], effects of radar signal penetration in the glacier surface due to different acquisition seasons and frequencies are not considered in this analysis, since the scope of this study is to analyze solely the error in elevation and volume changes using different void interpolation approaches. Thus, the resulting values are not suitable for, e.g., glacier mass balance computation without applying adequate corrections.

Two case studies were defined to carry out the analysis and artificial voids were generated in the respective elevation change fields.

The first case study aims to investigate the performance of the void filling approaches on large data voids. To this end, two subsets of the study region were selected, which cover different glacier types (see Figures 1 and 2). One covers wide sections of the Juneau Icefield (~869 km$^2$ glacier area) with a relatively smooth ice topography and mainly small elevation changes (called the "*Juneau*" setup in the following). The other covers mainly the RGI60-01.20686 glacier at the center of the study region, which expands from close to sea level up to ~2200 m a.s.l. over a ~142 km$^2$ ice area (called the "*Center*" setup in the following). At both setups, artificial voids were generated to simulate large data gaps. The strip-shaped voids (called "*Strip*" voids in the following) simulate gaps between satellite acquisitions. Cloud cover, low image contrast or sensor saturation can lead to large voids in the accumulation areas, when using optical satellite data to generate elevation information. The large circular voids (called "*Circle*" voids in the following) simulate these issues. Additionally, the complete absence of a glacier section is simulated at the Center setup. Therefore, the lower most section at the ablation area of the glacier was cut off (called the "*Terminus*" void in the following).

For the second case study, the whole elevation change field (Figure 1) was considered.

Advanced Spaceborne Thermal Emission and Reflection Radiometer (ASTER) stereo imagery is frequently used to compute geodetic glacier mass balances [5,21,22]. Low correlation values of the stereoscopic DEM processing indicate areas with less reliable elevation information or failures in the DEM reconstruction. With this motivation, a correlation mosaic obtained from 99 ASTER stereo images was used as a baseline for the artificial void generation (provided by R. McNabb; [9]). Following [9], a correlation threshold of 50% was used to produce an artificial void mask and to generate voids (19% void fraction) in the whole elevation change data set, called the "*Correlation*" setup in the following.

## 3.2. Void Filling

Three novel (two geometric synthesis and one compressed sensing-based approach) and three classical void filling techniques, recommended by [9], were applied to reconstruct glacier elevation change information in the artificially voided elevation change fields. Based on the concept (e.g., consideration of information around the void, smooth propagation and transition of the information at the void boundaries, continuity of isophotes), published results and the availability of implementations of the methods, we selected Telea and Navier–Stokes inpainting techniques, two geometric approaches, and shearlet inpainting, a compressed sensing-based approach. An overview of the used abbreviations for the different approaches and parametrizations is provided in Table 1. A detailed description of the implementation of the novel inpainting approaches in provided in the Supplement.

**Table 1.** Overview of applied void filling approaches and parametrizations.

| Approach | Parametrization | Abbreviation |
|---|---|---|
| *Novel Void Filling Approaches* | | |
| Telea | search radius | 2 — Telea-02 |
| | | 5 — Telea-05 |
| | | 8 — Telea-08 |
| | | 10 — Telea-10 |
| | | 15 — Telea-15 |
| | | 20 — Telea-20 |
| Navier–Stokes | search radius | 2 — NS-02 |
| | | 5 — NS-05 |
| | | 8 — NS-08 |
| | | 10 — NS-10 |
| | | 15 — NS-15 |
| | | 20 — NS-20 |
| Shearlet | nscales | 5 — SL-5 |
| | | 6 — SL-6 |
| | | 7 — SL-7 |
| *Classical Void Filling Approaches* | | |
| Bilinear | | CL-BL |
| Hypsometric | local | CL-LO |
| | global | CL-GL |

### 3.2.1. Telea Approach

This inpainting technique was developed to iteratively fill data voids [23]. It intends to mimic the work of professional image reconstructors by gradually propagating information from the boundary of the voids inwards. Thereby, the algorithm aims to approximate the intensity at point $p$ on the boundary $\delta\Omega$ of the void $\Omega$ by smoothly extrapolating the intensity $I$ from a known region $B_\varepsilon(p)$ around $p$ ($\varepsilon$: radius around $p$). For small $\varepsilon$, the extrapolation can be performed using the intensity gradient $\nabla I$ at point $q$ in $B_\varepsilon(p)$:

$$I_q(p) = I(q) + \nabla I(q)(p - q). \tag{1}$$

By adding up the estimates throughout $B_\varepsilon(p)$ and applying a weighting function $\omega(p,q)$, the intensity at point $p$ is obtained by

$$I(p) = \frac{\sum_{q \in B_\varepsilon(p)} \omega(p,q)[I(q) + \nabla I(q)(p-q)]}{\sum_{q \in B_\varepsilon(p)} \omega(p,q)}. \tag{2}$$

The exact form of the weighting function is fundamental for high-quality propagation of sharp image contours and smooth areas into the void areas. Telea [23] designed this function as a product of three components that are estimated for each point $p$ along the boundary of the void using the known pixel information within $B_\varepsilon(p)$. The directional component assigns a higher weight to the contribution from the pixels next to the normal direction of the boundary at $p$. The geometric distance component ensures a gradual decrease in the contribution from the pixels further away from $p$. The level set distance component leads to a stronger contribution from the pixel next to the contour through $p$. The inpainting is iteratively performed for all pixels at the boundary of the void, and by propagating the boundary into the data void until it is completely inpainted. By starting the inpainting process at the boundary of the void and gradually moving into the void, the algorithm mimics the process used by human image reconstructors [14]. To carry out this propagation, the fast marching method (FMM) [24] is used. It solves the Eikonal equation:

$$|\nabla T| = 1 \text{ on } \Omega, \quad T = 0 \text{ on } \delta\Omega \tag{3}$$

where solution $T$ is the vector field such that for all pixels in the void, $p$ in $\Omega$, $T(p)$ is the distance to the boundary of the void $\delta\Omega$. The FFM ensures that the pixels are inpainted in the order of increasing distance to $\delta\Omega$, i.e., increasing $T$ [25].

A more detailed description of the inpainting approach and the implementation can be found in [23].

### 3.2.2. Navier–Stokes Approach

Bertalmio et al. [13] suggested an inpainting technique that uses the Navier–Stokes equation to fill voids in images based on the information from the surrounding regions. Specifically, classical fluid dynamics are applied to advance isophotes (lines of equal image intensity) into the void regions. Image intensity $I$ is considered as a stream function (defined below) of a 2D incompressible flow. The direction of the perpendicular gradient $\nabla_p I$, where $\nabla_p = \left(-\delta_y, \delta_x\right)$, describes the isophote lines that must be parallel to the level curves of $\Delta I$, the Laplacian of the image, which is an estimator of the image smoothness. This condition leads to

$$\left(\nabla_p I\right) \cdot \left(\nabla(\Delta I)\right) = 0 \tag{4}$$

where the dot stands for the inner product between two vectors.

Equation (4) can be seen as a transport equation of the image intensity $I$ along the level curves of $\Delta I$. To understand this, consider a 2D incompressible Newtonian fluid. The vorticity–stream function formulation for the Navier–Stokes equations is

$$\delta_t \omega + v \cdot \nabla \omega = \nu \Delta \omega \tag{5}$$

where $\omega = \nabla \times v$ is the vorticity, $v$ is the fluid velocity and $\times$ stands for the cross product between two vectors. Introducing the stream function $\Psi$ that satisfies $\nabla_p \Psi = v$, for very low viscosity $\nu$, we obtain the following steady-state solution of Equation (5):

$$\left(\nabla_p \Psi\right) \cdot \left(\nabla(\Delta \Psi)\right) \approx 0 \tag{6}$$

which is identical to Equation (4) if Ψ is replaced by *I*. Consequently, to solve the inpainting Equation (4), it is sufficient to find a steady-state stream function Ψ. The advantage of this approach is that it can rely on existing theoretical and numerical solutions of the Navier–Stokes problem.

The correspondence between the variables for a 2D incompressible fluid flow problem and the image inpainting problem is summarized in Table 2.

**Table 2.** Correspondence between the variables of an incompressible fluid flow problem and the Navier–Stokes inpainting problem.

| Fluid Dynamics | Inpainting |
|---|---|
| Stream function Ψ | Image intensity *I* |
| Fluid velocity $v = \nabla_p \Psi$ | Isophote direction $\nabla_p I$ |
| Vorticity $\omega = \Delta \psi$ | Image smoothness $\Delta I$ |
| Viscosity $v$ | Anisotropic diffusion $v$ |

Low-viscosity fluid problems are likely to have long relaxation times to reach steady state. Therefore, an anisotropic diffusion term that preserves steep gradients is added to Equation (4); the term is large for smooth areas and is close to zero for steep gradients.

Similar to the Telea approach, the boundary conditions (derivatives of *I*) at the boundary of the voids are derived from the known pixel intensity information within a predefined search radius. The detailed mathematical background and more information on the implementation of this image inpainting approach can be found in [13].

### 3.2.3. Shearlet Approach

The shearlet approach is based on compressed sensing, an area of signal processing that started from the works of Donoho and Candès, Romberg and Tao in 2004 [26,27]. The underlying principle of compressed sensing is that the compressibility of structured data makes it possible to reconstruct the data from a highly incomplete set of linear measurements. Natural images are highly compressible in the sense that most of their energy is concentrated in a small subset of coefficients when representing them in a suitable basis or (more generally) a tight frame. The basis or frame that satisfies this property is called the sparsifying representation. The sparsifying representation we use is based on shearlets, a structured system of functions that are constructed by applying translation, scaling and shearing operators to a prototype function. For the readers familiar with wavelets, shearlets are similar in spirit, but form a richer system in which many important classes of images are much more sparse. For more details on shearlets, we refer to [28]. When constructing the shearlet system, it is important to choose the scale of the basis elements to be large enough, such that the length of the largest basis elements are at least as large as the size of the largest void.

In the inpainting problem, we assume that the vector of shearlet coefficients $c^*$ corresponding to the true (vectorized) image *I* is sparse, meaning that most of its elements are zero and only a few are nonzero. Under this assumption, the compressed sensing paradigm tells us that it is natural to try to estimate $c^*$ by solving the following optimization problem:

$$\hat{c} = \underset{c}{argmin} c_0 \text{ such that } P\Phi c = PI. \tag{7}$$

Above, *P* denotes the orthogonal projection of an image onto the set of known pixels, i.e., *PI* is the image in which the unknown part is set to zero; Φ is the synthesis shearlet operator that converts shearlet coefficients to images; and $\|c\|_0$ denotes the number of nonzero coefficients in *c*, also known as $l_0$-norm. The meaning of this optimization problem is the following: we are trying to find the sparsest vector of shearlet coefficients, $\hat{c}$, that corresponds to an image, $\Phi\hat{c}$, that agrees with the measured data, *PI* (the non-void pixels). Once $\hat{c}$ is found, an estimate for the true image may be easily obtained by taking the inverse shearlet transform $\hat{I} = \Phi\hat{c}$.

Unfortunately, the optimization problem in Equation (7) is known to be NP–hard (non-deterministic polynomial-time hard) and, therefore, may not be solved efficiently. There are two powerful approaches to deal with this difficulty: (i) apply convex relaxation by substituting the $l_0$-norm of $c$ in Equation (7) with the $l_1$-norm $\|c\|_1 = |c_1| + |c_2| + \ldots$; or (ii) apply an iterative algorithm. Both methods are principled in the sense that there are mathematical theorems that guarantee that, under suitable assumptions, they provide the exact solution to Equation (7).

Following the recommendations from [17], we applied an iterative thresholding algorithm with a decaying threshold. The update rule of the algorithm is

$$\hat{I}^{n+1} = \Phi[H_{\lambda_n}\Phi^*[\hat{I}^n + P(I^0 - \hat{I}^n)]]. \tag{8}$$

Above, $n \in \{0, \ldots, N-1\}$ is the iteration index; $\hat{I}^n$ denotes the inpainted image after the $n$-th update; the initial estimate $\hat{I}^0$ is set to an all zeros vector; $I^0 = PI$ denotes the masked image, i.e., the measured image with the unknown pixels set to zero; $\Phi^*$ denotes the analysis operator of the shearlet transform, i.e., the operator that converts images to shearlet coefficients; and $H_{\lambda_n}$ is the element-wise hard-thresholding operator defined as

$$H_\lambda(y) = \begin{cases} y, \text{if } |y| \geq \lambda \\ 0, \text{ otherwise.} \end{cases} \tag{9}$$

The threshold $\lambda_n$ is set to decay exponentially with the iteration number, $n$, according to

$$\lambda_n = \frac{\delta\alpha^n}{N-1} \tag{10}$$

where $\delta$ is the absolute value of the largest coefficient of $PI$ in the shearlet domain; and $\alpha$ is a parameter of the algorithm that may be adjusted to optimize performance.

In simple terms, the algorithm may be described as follows. At the first iteration, the shearlet transform $\Phi^*$ of the masked image $I^0$ is calculated, the coefficients that are smaller than the threshold are set to zero (this ensures that the solution has sparse shearlet coefficients) and the inpainted image $\hat{I}^1$ is obtained from the sparse shearlet coefficients via the inverse shearlet transform. At each of the remaining iterations, the following operations are applied consecutively. First, the inpainted image is set to the measurement values at the indices associated with the known pixels (this is implemented via $\hat{I}^n + P(I^0 - \hat{I}^n)$ in Equation (4)).

Then, just as before, the shearlet transform $\Phi^*$ of the result is calculated, the coefficients that are smaller than the threshold are set to zero (ensuring sparsity) and the next update to the inpainted images is obtained from the sparse shearlet coefficients via the inverse shearlet transform. The parameter $\alpha$ controls the threshold and thus the sparsity of the solution. The number of iterations $N$ should be chosen as high enough to ensure convergence.

### 3.2.4. Classical Void Filling

Three classical void filling techniques were applied for comparison with the novel inpainting methods. We selected three recommended approaches by McNabb et al. [9].

The bilinear (BL) interpolation was carried out using the r.fillnulls module (default settings except method = bilinear) of the Geographic Resources Analysis Support System (GRASS) software, Version 7.4 (https://grass.osgeo.org).

The hypsometric approaches were carried out according to [9,20]. For the local hypsometric approach (LO), the average elevation change of individual glaciers was calculated by using elevation bins of 10% of the glacier elevation range, if it is <500 m, and bins of 50 m for glaciers with elevation ranges of >500 m. Voids in the hypsometric elevation change distribution are filled by applying a third-order polynomial fit. For the global hypsometric approach (GL), 50 m elevation bins were used to compute regional mean values. Outliers in the respective elevation bins (global and local analysis)

are filtered out using a 298–% quantile filter. The SRTM DEM is used as elevation reference for the hypsometric analyses. Only the global hypsometric approach was applied at the Juneau and Center setups, since the setups are relatively small and cover mainly just one ice body. The hypsometric void filling was implemented in R.

### 3.3. Error Metrics and Comparison Methodology

#### 3.3.1. Large Voids—Juneau and Center Setups

The Juneau and Center setups were used to study the performance of the different void filling approaches for specific large voids. The difference $dh_i = dh_i^{IP} - dh_i^{O}$ of the inpainted $dh_i^{IP}$ and original $dh_i^{O}$ glacier elevation change data on the void areas was calculated for each individual pixel $i$. Further, the relative error $dh_i^r = dh_i/dh_i^{O}$ was computed.

In addition to pixel-wise error metrics, we are also interested in global error metrics. One important quantity is the mean relative offset:

$$\overline{dh}^r = \frac{\sum_{i=1}^{N} dh_i}{\sum_{i=1}^{N} dh_i^{O}} \tag{11}$$

where $N$ is the total number of inpainted pixels in a region of interest. It quantifies the average error in glacier mass change estimation introduced by the void filling in relation to the original data. It is also a measure for a potential general bias generated by the void filling, since local or pixel-wise variations will average out.

A variability in the elevation change offsets on small spatial scales is visible, in particular at the Center void at the Juneau setup (Figure 2, right panel). Such high-frequency variations of the glacier elevation changes are caused by limitations of the underlying data sets (noise in the DEMs) and will cancel out when integrating over the glacier area in volume change computations. The reconstruction of such high-frequency variations is beyond the scope of the void filling algorithms and, in fact, is neither desirable nor possible. However, these high-frequency variations can lead to large errors when analyzing the quality of the algorithms on small spatial scales and consequently hamper the analysis of the void filling results. The impact of the high-frequency noise on the analysis was suppressed by applying 2D Gaussian smoothing on the difference maps *dh*,

$$dh_i^f = [dh * g]_i \tag{12}$$

where $g$ is the 2D Gaussian kernel with a standard deviation of ten pixels (300 m, half of the maximum correlation length for the tested algorithms, see Section 4.2.3) in each direction. Intuitively, this says that we only require the algorithms to be accurate on the scales larger than 300 m.

Subsequently, averaging the absolute value of the filtered per pixel error $dh_i^f$ leads to the filtered average absolute error metric

$$AAE^f = \frac{\sum_{i=1}^{N} \left| dh_i^f \right|}{N} \tag{13}$$

This metric is a measure for the average absolute error generated by the void filling on pixel scales, since positive and negative contributions to the error in different pixels will not cancel out.

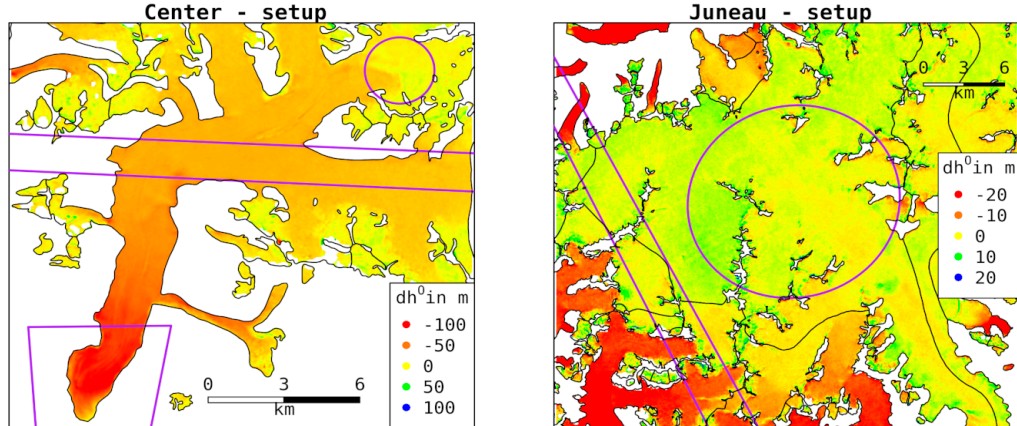

**Figure 2.** Original elevation change fields of subsets of the study region (Figure 1) used for analysis of the void filling approaches on large data voids (purple polygons). Results of the different void filling approaches are provided in Figure 4, Figures S1 and S2.

### 3.3.2. Large Region—Correlation Setup

To analyze the impact of the void filling methods on large spatial scales, we used the Correlation setup. For mass balance computations, it is important to have information on the mean offset $\overline{dh}$ generated by the void filling:

$$\overline{dh} = \frac{\sum_{i=1}^{N} dh_i}{N}.$$ (14)

In order to infer differences between the methods, we computed conservative, simultaneous confidence intervals for $\overline{dh}$ by setting $\alpha = 0.05/18^2$ (18 void filling methods were tested). Similarly, we also computed the standard deviation $\sigma_{dh}$ of $dh_i$ and the respective simultaneous confidence intervals for $\sigma_{dh}$. The methods can be compared for statistically significant differences by looking for overlap of the simultaneous confidence intervals.

Elevation changes are, in general, affected by spatial auto-correlation. Correlation lengths $d_{cor}$ for the void filling offsets ranging between 3 and 20 pixels (90–600 m, see Table 3) are obtained for the different methods based on semivariogram analysis. In order to compute correct confidence intervals for $\overline{dh}$ and $\sigma_{dh}$, we should have independent samples $dh_i$. To guarantee the independence of $dh_i$, we accounted for the spatial auto-correlation and performed a regular subsampling of $dh_i$ with a distance of $d_{cor}$ to the next void-filled pixel.

### 3.3.3. Impact of Void Filling on Different Scales

Besides estimating the glacier volume changes on regional scales, it is also important to generate such information on different sub-scales, e.g., on glacier scales as requested by the World Glacier Monitoring Service. Let $\overline{dh}_{gl}$ be the mean elevation change offset on the void area in a particular glacier area; this quantity is defined exactly as in Equation (14) with the only difference that the summation is over the void pixels in the glacier area of interest. The central question is how can we estimate $\overline{dh}_{gl}$ for glacier areas for which we do not have the ground truth data? The glacier area sizes and void sizes and fractions can vary strongly from study site to study site, affecting $\overline{dh}_{gl}$. In order to estimate the impact of the void filling on volume change computations at different scales, we propose the following approximate relation:

$$\overline{dh}_{gl} = \overline{dh} \pm C \frac{\sigma_{dh}}{\sqrt{N}}$$ (15)

where $\overline{dh}$ and $\sigma_{dh}$ are obtained in Section 3.3.2 (Table 3), $N$ is the number of void pixels in the glacier area and $C$ is a factor that, as explained in Section 4.2.3 below, depends on $d_{cor}$ and hence is (somewhat) different for each void filling approach. Theoretically, Equation (15) is a consequence of the central limit

theorem. Each reconstruction algorithm will have a bias, estimated by $\overline{dh}$, which will be the dominant source of error in very large regions (when $N$ is large). It will also have a variance, estimated by $\sigma_{dh}$, which will dominate the error in each particular reconstructed pixel. Since the errors in different pixels are independent (as long as the pixels are separated by about $d_{cor}$), by the central limit theorem, due to averaging of the errors, on the glacier scale, the contribution from variance will decrease as $1/\sqrt{N}$.

The Correlation setup was taken to test the proposed relation. We selected the best performing approaches (shearlet with nscales = 5 and bilinear, see Section 4.2.2). Hypsometric void filling is widely used in mass balance studies (e.g., [5,6,20]), thus we also included the local hypsometric approach in our analysis. To test Equation (15) on different scales, we selected all glaciers with a <40% void fraction (two times the void fraction of the Correlation setup taken as a whole), covering 76% of the Correlation setup (378 glaciers). For glaciers with a large void fraction (>40%), it seems difficult to carry out an accurate reconstruction by most of the methods we considered and thus we assume that our proposed approximation is not valid anymore. For each glacier, the void pixel count $N$ and the average offsets from the original data $\overline{dh}_{gl}$ were calculated. Subsequently, various analyses were carried out to evaluate the proposed relation; the results are reported in Section 4.2.3.

**Table 3.** Mean $\overline{dh}$ and standard deviation $\sigma_{dh}$ of sampled offset values for the Correlation setup including lower, $\overline{dh}^L$, and upper, $\overline{dh}^U$, bounds of simultaneous 95% confidence intervals. $d_{cor}$: correlation length of inpainted offsets.

| Approach | $\overline{dh}$ (m) | $\overline{dh}^L$ (m) | $\overline{dh}^U$ (m) | $\sigma_{dh}$ (m) | $\sigma_{dh}^L$ (m) | $\sigma_{dh}^U$ (m) | $d_{cor}$ (pixel) |
|---|---|---|---|---|---|---|---|
| NS-02 | −0.0538 | −0.0947 | −0.0128 | 2.4515 | 2.4286 | 2.4748 | 5 |
| NS-05 | −0.0503 | −0.0931 | −0.0074 | 2.5643 | 2.5403 | 2.5887 | 5 |
| NS-08 | −0.0497 | −0.0928 | −0.0066 | 2.5811 | 2.5570 | 2.6057 | 5 |
| NS-10 | −0.0496 | −0.0928 | −0.0064 | 2.5876 | 2.5634 | 2.6122 | 5 |
| NS-15 | −0.0489 | −0.0922 | −0.0056 | 2.5943 | 2.5700 | 2.6189 | 5 |
| NS-20 | −0.0481 | −0.0915 | −0.0047 | 2.5990 | 2.5747 | 2.6237 | 5 |
| Telea-02 | −0.0480 | −0.0901 | −0.0058 | 2.5222 | 2.4986 | 2.5461 | 5 |
| Telea-05 | −0.0758 | −0.1315 | −0.0200 | 2.7893 | 2.7632 | 2.8158 | 6 |
| Telea-08 | −0.0774 | −0.1377 | −0.0171 | 3.0168 | 2.9886 | 3.0455 | 6 |
| Telea-10 | −0.0773 | −0.1400 | −0.0145 | 3.1387 | 3.1093 | 3.1685 | 6 |
| Telea-15 | −0.0632 | −0.1548 | 0.0284 | 3.4256 | 3.3936 | 3.4582 | 8 |
| Telea-20 | −0.0521 | −0.1469 | 0.0427 | 3.5475 | 3.5143 | 3.5812 | 8 |
| SL-5 | −0.0090 | −0.0354 | 0.0175 | 1.9837 | 1.9651 | 2.0025 | 4 |
| SL-6 | −0.0239 | −0.0518 | 0.0039 | 2.0891 | 2.0695 | 2.1089 | 4 |
| SL-7 | −0.0113 | −0.0520 | 0.0293 | 2.4329 | 2.4101 | 2.4560 | 5 |
| CL-BL | −0.0054 | −0.0334 | 0.0226 | 2.1007 | 2.0811 | 2.1207 | 4 |
| CL-GL | −0.3240 | −0.7269 | 0.0788 | 12.1213 | 12.0080 | 12.2365 | 10 |
| CL-LO | −0.0440 | −0.2474 | 0.1594 | 6.1212 | 6.0640 | 6.1794 | 10 |

## 4. Results and Discussion

### 4.1. Void-Free Volume Changes

For the total analyzed glacier area of 6210 km$^2$, a void-free volume change of −52.9 ± 11.9 km$^3$ (corresponding to an average elevation change of −8.52 ± 1.92 m) was found for the period 2000–2012/13 (based on the data sets provided by R. McNabb; [9]). The hypsometric distribution of the elevation changes binned in 50 m intervals is plotted in Figure 3. Average surface lowering of up to ~35 m is obtained for the lower-elevation intervals. Above ~1300 m a.s.l., a positive surface elevation change is found. The contribution of glacier sections above 2000 m a.s.l. to the total volume change is minimal, due to the small glacier area.

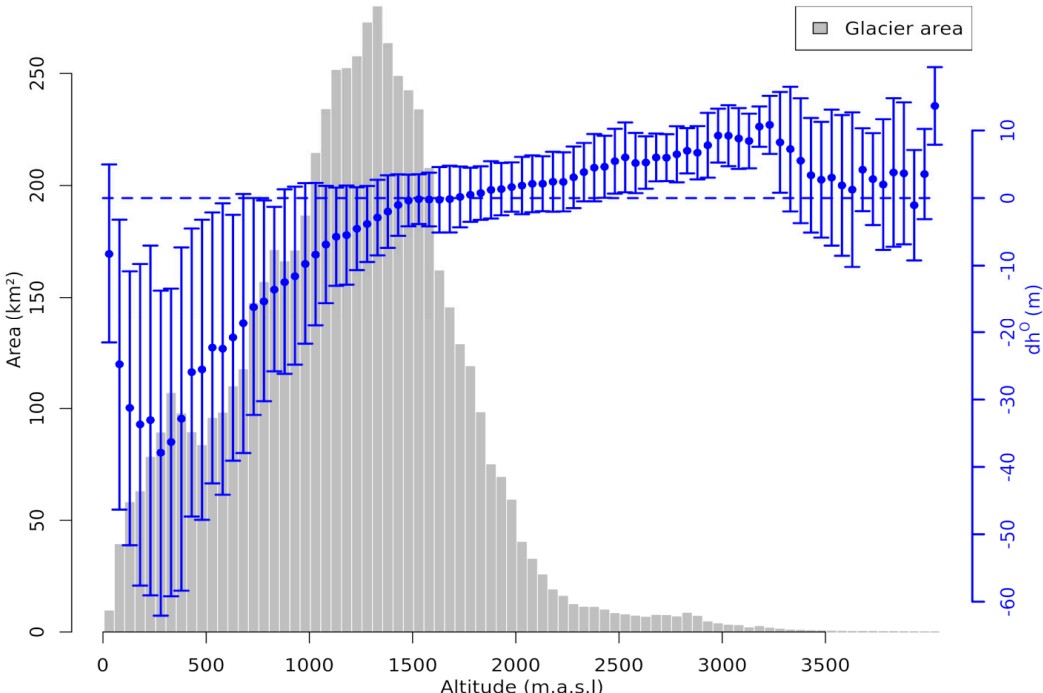

**Figure 3.** Hypsometric distribution of glacier area (gray bars) and surface elevation changes $dh^O$ (blue dots: mean $dh^O$, error bars: median absolute deviation of $dh^O$) of the complete void-free data set (Correlation setup), grouped in 50 m altitude bins.

*4.2. Evaluation of Void-Filled Data Sets*

4.2.1. Large Voids—Juneau and Center Setup

In Figure 4, for the Center and Juneau setups, we present one inpainting result per approach. For other parametrizations, the inpainting results are displayed in Figures S1 and S2. The values of $dh$ and $dh^r$ are illustrated in Figures S1 and S2 for the tested methods at both setups. The values of $\overline{dh}^r$ and $AAE^f$ for the individual voids at both setups are provided in Figure 5. First, based on the visual inspection of the reconstructed data in Figures S1 and S2, we conclude that the shearlet approach with nscales = 5 did not lead to adequate reconstructions of the voids in the Juneau and Center setups. The size of shearlet basis functions with nscales = 5 is too small for such large voids. As a result, away from the void boundary, a nearly constant (zero) function was inpainted. For some sub-regions, this leads to small $\overline{dh}^r$ and $AAE^f$ values by chance, but in general, the performance is bad. These values will mislead the method comparison and we consequently removed the shearlet approach with nscales = 5 from the further discussion of the Juneau and Center setups analysis.

**Center setup:** Telea and Navier–Stokes achieve reasonable $\overline{dh}^r$ and $AAE^f$ values for all voids. The shearlet inpainting led to considerably larger errors (in terms of $\overline{dh}^r$) at the Circle and Terminus voids. The hypsometric approach shows the lowest $AAE^f$ and a small $\overline{dh}^r$ at the Circle and Terminus voids. The visual inspection of its reconstruction results (Figure S1) also suggests the conclusion that the hypsometric approach leads to the best reconstruction (among all methods) in these two regions. However, at the Strip void, the hypsometric approach shows much higher $\overline{dh}^r$ and $AAE^f$ values than the competing methods.

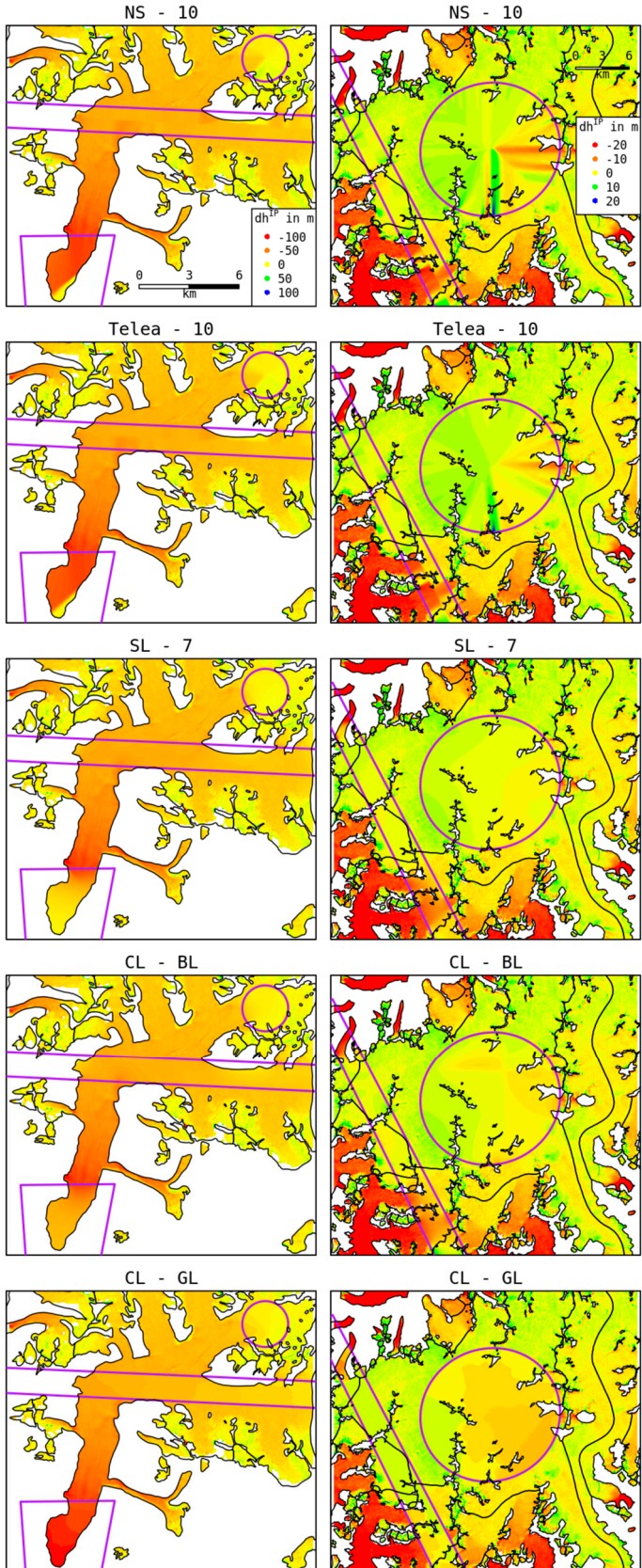

**Figure 4.** Void filling results of the different approaches (one per approach) at the Center (left column) and Juneau (right column) setups. See Figures S1 and S2 for more results.

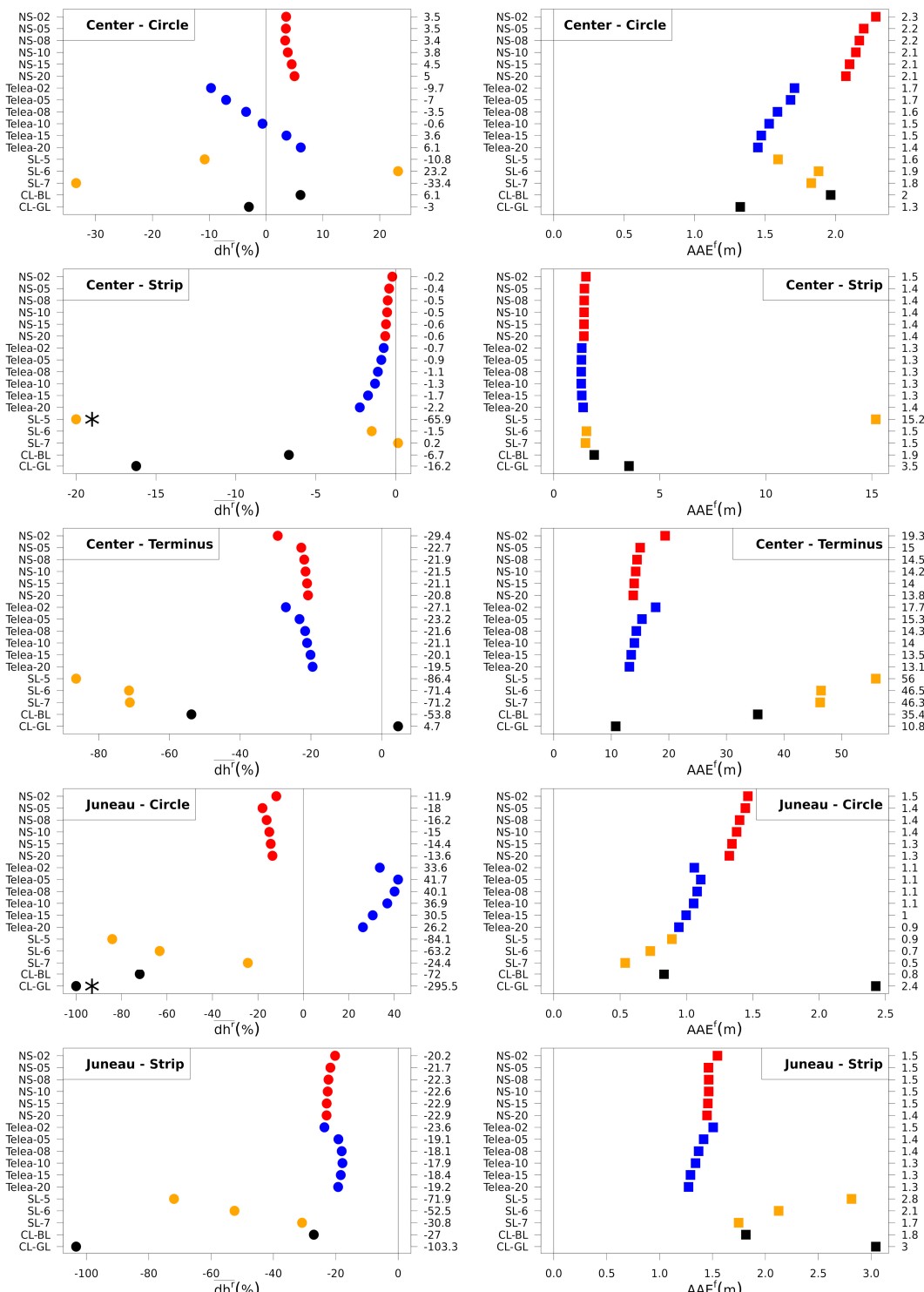

**Figure 5.** Mean relative offset $\overline{dh}^{r}$ and absolute average error $AAE^{f}$ of the individual voids for the Center and Juneau setups. * values out of bounds.

The Center setup consists mainly of one glacier with a highly variable elevation change pattern that strongly depends on the glacier elevation (Figure S3). The Circle void is situated in the upper reaches of the accumulation area of the glacier, with a moderate elevation change signal. There are data nearly all around the void to constrain the Telea, Navier–Stokes and bilinear inpainting methods, leading to the good performance of these approaches. However, at the Terminus void, the bilinear approach is not able to reconstruct the original data adequately. There are only data available on one

side of this void, which is insufficient to constrain the bilinear void filling. The same effect explains the bad performance of the shearlet approach at the Terminus void. The Telea and Navier–Stokes approaches show a better performance, since both techniques attempt to reconstruct the data by incorporating information on the elevation change gradient from the void boundary. This is especially important, because at the Terminus void the strongest elevation changes are found.

At the Terminus and Circle voids, there is sufficient coverage of the relevant elevation bins for the hypsometric approach, allowing a good reconstruction of the original data. Note that the good reconstruction at the Terminus void by the hypsometric approach is based on the extrapolation of the hypsometric elevation change distribution (Figure S3) of the known data using a third-order polynomial fit in order to fill the completely voided elevation bins of the Terminus void. Here, the elevation change pattern followed the trend of the elevation bins further upwards. This is not always the case, so for other glaciers and especially glacier types, this might lead to a failure; for example, this happens at the Juneau Strip setup as discussed below.

**Juneau setup:** At the Circle void, the Navier–Stokes approach shows the smallest $\overline{dh}^r$ value. Telea and shearlet (nscales = 7) inpainting methods also show a good performance. The lowest $AAE^f$ is found for the shearlet and bilinear void filling methods. At the Strip void, Telea and Navier–Stokes approaches lead to the smallest $\overline{dh}^r$ and $AAE^f$ values. The hypsometric approach shows the worst performance for both voids and metrics.

The Juneau setup is dominated by the accumulation area of the Juneau Icefield, with a smooth topography and relatively small elevation changes, where the Circle void is located. There is sufficient data around the void with a similar elevation change pattern to constrain the shearlet inpainting for the Circle void, leading to the good performance of the shearlet approach in contrast to its performance at the Center setup. The voids at the Juneau setup are also wider than those of the Center setup. Consequently, shearlet inpainting with a larger base function (nscales = 7) generated better results.

The Strip void covers sections of the outlet glaciers of the Juneau Icefield, with more heterogeneous elevation change patterns. Consequently, the Telea and Navier–Stokes approaches, which are constrained only by values next to the void boundaries, show the best performance here. There are discontinuities in the inpainted images produced by the Telea and Navier–Stokes methods in particular at the very large Circle void; shearlet and bilinear void filling generates smooth results. For large inpainting radii, the reconstructions by the Telea and Navier–Stokes approaches are getting smoother. The bad performance of the hypsometric approach can be attributed to the fact that the information for the respective elevation bins is gathered from the whole setup, not reflecting spatial differences within the elevation bins, due to different glacier settings (e.g., aspect).

**Summary:** The findings of the case study lead to the following conclusions.

In order to obtain good reconstructions, the size of the shearlet base function must be sufficiently large compared to the void size; this explains the failure of shearlet with nscales = 5. Further, the shearlet approach needs sufficient known data from all around the void to be adequately constrained; the lack of such data explains the bad performance of shearlets at the Center Terminus void.

The hypsometric approach sometimes generates good results when applied to a single glacier (Center setup) but can be unstable when applied to a wider data set.

The bilinear method, similar to shearlet, needs sufficient known data from all around the void to produce good results. The lack of such data explains the bad performance at the Center Terminus void. In general, the bilinear method showed no outstanding performance for the relatively large voids, especially when pronounced gradients needed to be reconstructed, like for the Center setup.

Both the Navier–Stokes and Telea approaches show an all-around good performance for all the different voids. Compared to other methods, these two methods are more stable. In all cases and for both metrics, the performance is either the best or not far from the best. In terms of $\overline{dh}^r$, the Navier–Stokes approach outperforms the Telea approach at the Juneau Circle void. For both methods, generally smaller $AAE^f$ values are found for larger inpainting radii but no clear relation between $\overline{dh}^r$ and the inpainting radius is seen.

### 4.2.2. Large Region—Correlation Setup

The values of $\overline{dh}$ and $\sigma_{dh}$ including the corresponding upper and lower limits of the *simultaneous* 95% confidence intervals are provided in Table 3 and displayed in Figure 6. To construct the conservative simultaneous confidence intervals, we constructed the *individual* confidence intervals with a significance level of $\alpha = 0.05/18^2$, because 18 void filling methods were tested. The simultaneous confidence intervals of $\overline{dh}$ overlap for all methods (number of samples is different for each method and ranges from 6485 to 80,540, depending on the spatial autocorrelation, see Section 4.2.2). Thus, the differences in $\overline{dh}$ are not statistically significant for all methods. For $\sigma_{dh}$, there exist statistically significant differences.

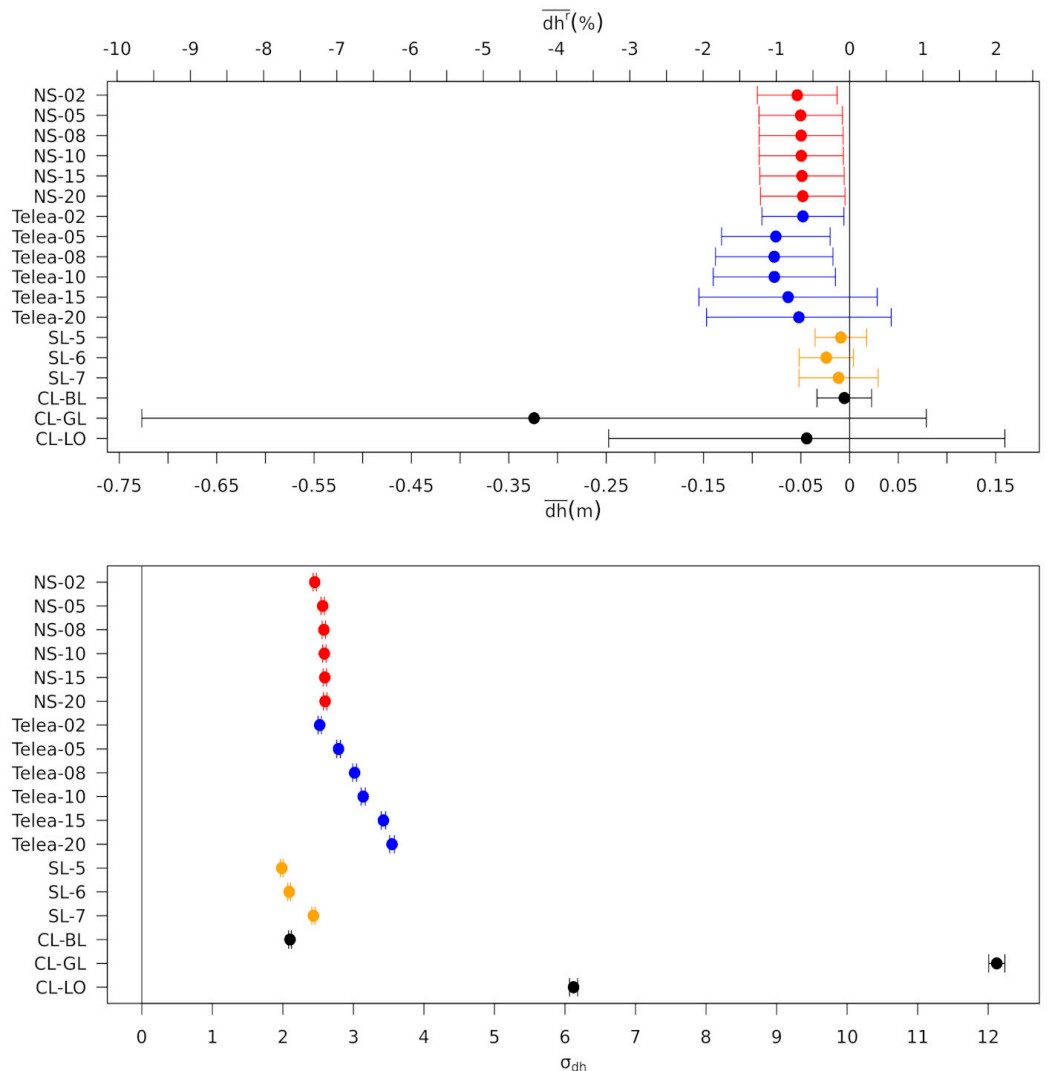

**Figure 6.** Mean $\overline{dh}$ ($\overline{dh}^r$: relative offset to original data) and standard deviation $\sigma_{dh}$ of sampled offset values for the Correlation setup including simultaneous confidence intervals.

The global hypsometric approach shows the statistically significantly largest $\sigma_{dh}$ value, followed by the local hypsometric method, indicating that the largest pixel-wise offsets are caused by these methods. In Figure 6, we also display the values of the relative mean offsets $\overline{dh}^r$ (upper x-axis) for the Correlation setup, to get an impression of the overall error caused by the void filling. In general, $\overline{dh}^r$ is well below ±2.5% (considering the confidence intervals) for all methods, except for the hypsometric approaches. The lower bound of the confidence interval for $\overline{dh}^r$ is close to −10% for the hypsometric global approach, which makes us assume that considerable offsets may be introduced by the hypsometric approaches, or at least we cannot guarantee that this will not happen based on the available data. The above

findings can be attributed to the nature of the hypsometric approaches, using average values to fill voids in certain elevation intervals. In particular, the global hypsometric approach, which uses the average value of the whole study region, causes larger $\sigma_{dh}$ and also potentially large (in absolute value) $\overline{dh}$ and $\overline{dh}^r$, since spatial differences in the elevation changes throughout the study region are disregarded. This fact explains also the better performance of the local hypsometric approach, in the sense of a tighter confidence interval, which guarantees that $\overline{dh}$ and $\overline{dh}^r$ are small, since the void filling is based only on information aggregated on glacier scales (see also Section 4.2.1).

For the Navier–Stokes approach, both $\sigma_{dh}$ and $\overline{dh}$ show nearly no dependence on the inpainting radius used. The confidence intervals of $\overline{dh}$ and $\overline{dh}^r$ do not overlap with zero. Thus, it is likely that this method will generate a bias, even though the bias will be reasonably small (confidence intervals for $\overline{dh}$ range between −0.0947 and −0.0128).

For the Telea approach, $\sigma_{dh}$ increases significantly with the inpainting radius. For Telea-20, $\sigma_{dh}$ is about 1.4 times larger than for Telea-02. This means Telea with a large search radius introduces much larger pixel-wise errors than Telea with a small search radius, and also larger than shearlets, Navier–Stokes and bilinear interpolation. Increasing $\sigma_{dh}$ also implies that the confidence interval for $\overline{dh}$ increases with the inpainting radius. For a large inpainting radius, the confidence interval for $\overline{dh}$ overlaps with zero, but it is also wider than the confidence interval for $\overline{dh}$ with a small search radius, so there is more uncertainty regarding where the true bias is. Still, even for Telea-15, the largest deviation of the confidence interval bound for $\overline{dh}$ from zero is −0.1548, which is reasonably small.

The statistically significantly smallest $\sigma_{dh}$ is generated by shearlet (nscales = 5 and 6) and by bilinear void filling. The simultaneous confidence intervals for $\overline{dh}$ of the bilinear and shearlet methods include zero, indicating that there is no statistically significant bias for these methods. For Telea (inpainting radii 15 and 20) and both hypsometric approaches, this is also the case. However, these methods have the widest confidence intervals, more so for the global hypsometric method, whereas the bilinear and shearlet methods have the overall smallest deviation of the limits of the confidence intervals from zero. Therefore, for bilinear and shearlet methods, we are very confident that the bias that these methods generate is very small.

As discovered in Section 4.2.1, a sufficiently wide margin with known data around the void is needed to well constrain the shearlet approach. In contrast to the Center setup, in the Correlation setup, this is the case for most voids (see Figure 1). The maximum distance of the void pixels to the next known pixel is up to 800 m for the Correlation setup, which is much smaller than that for the Center (up to 4 km) or Juneau (up to 8 km) setups. Consequently, higher performance is found for the shearlet approach when using smaller nscales values (base function length), allowing a better reconstruction of small-scale patterns. Further, for the bilinear approach, the void setting is favorable and contributes to its good performance.

Summing up the findings, based on the narrow confidence intervals that are nearly centered around zero, we can guarantee that the bilinear and shearlet (in particular for nscales = 5) void filling approaches generate small overall bias $\overline{dh}$ and $\overline{dh}^r$ and have statistically significantly the smallest $\sigma_{dh}$. The guarantees for other methods are weaker. In this sense, we consider the performance of bilinear and shearlet to be the best on the large Correlation setup among all methods (based on the data available to us). The Telea and Navier–Stokes and local hypsometric methods showed reasonable performance. Since the confidence intervals for the global hypsometric approach are so wide, we cannot guarantee that this method performs well, and since it has the highest $\sigma_{dh}$ among all methods, its performance on individual pixels is by far the worst. We believe that the global hypsometric approach is not suitable for accurate void filling. This method may still be useful when the voids are so large that there is no meaningful data to constrain other methods.

### 4.2.3. Impact of Void Filling on Different Scales

Based on Equation (15), we plotted $\overline{dh_{gl}} - \overline{dh}$ as a function of $\sigma_{dh}/\sqrt{N}$ in Figure 7 for the shearlet (nscales = 5), bilinear and local hypsometric void filling methods. As expected from Equation (15),

the values of $\overline{dh_{gl}} - \overline{dh}$ converge towards zero for large $N$ (small $\sigma_{dh}/\sqrt{N}$), but for small $N$, the distribution of $\overline{dh_{gl}} - \overline{dh}$ is rather uniform. Since the elevation offsets are spatially auto-correlated, we should replace $N$ with $N/d_{cor}^2$ ($d_{cor}^2$ is the size of the correlated group of pixels) so that Equation (15) becomes

$$\overline{dh_{gl}} = \overline{dh} \pm 2d_{cor}\frac{\sigma_{dh}}{\sqrt{N}} \text{ for } N \gg d_{cor}^2. \tag{16}$$

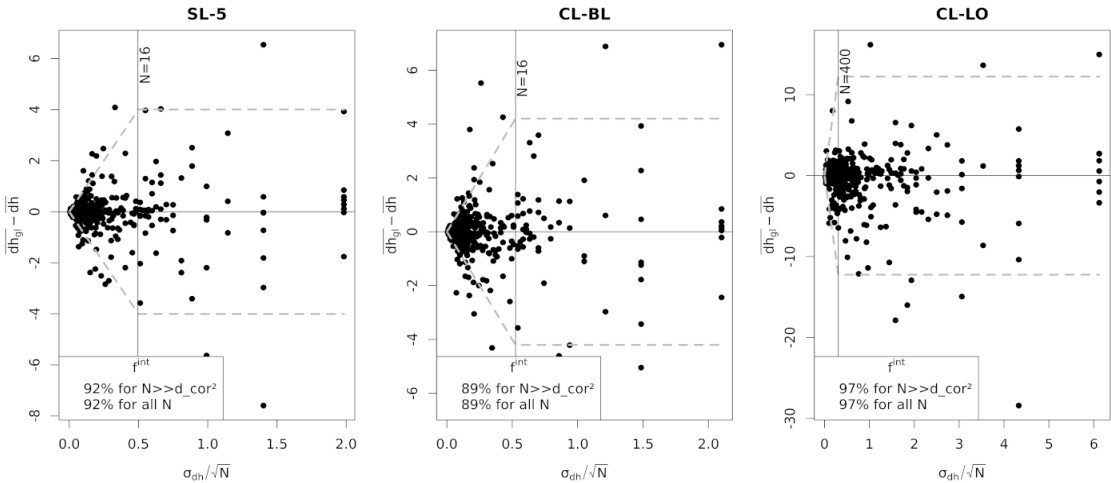

**Figure 7.** Scatter plot of $\left(\overline{dh_{gl}} - \overline{dh}\right)$ versus $\sigma_{dh}/\sqrt{N}$ for the three tested void filling approaches. Black vertical lines indicate correlation area approximation in pixels ($d_{cor}^2$). Gray dashed lines indicate the bounds from Equations (16) and (18) approximated by $2d_{cor}\sigma_{dh}/\sqrt{N}$ for $N > d_{cor}^2$ and by $2\sigma_{dh}$ for $N < d_{cor}^2$. $f^{int}$: fraction of points within the bounds.

Note that we have introduced a factor of 2 above, which corresponds to two standard deviations so that the interval in Equation (16) is the 95% confidence interval for $\overline{dh_{gl}}$ based on Normal approximation for the distribution of $\overline{dh_{gl}}$. We expect that the Normal approximation is valid when $N$ is large. Consequently, by the central limit theorem, the distribution of $X = \left(\overline{dh_{gl}} - \overline{dh}\right)\sqrt{N}$ must have a mean of zero and should be approximately Normal for $N \gg d_{cor}^2$ (concretely, $N = 10d_{cor}^2$ was applied as the lower threshold of $N$ in the subsequent analysis). We performed a two-sided t-test with the null hypothesis: mean of $X$ is zero. The resulting mean and *p*-values are summarized in Table 4.

**Table 4.** Summary of mean and standard deviation of X, *p*-value of t-test and revealed value of $\frac{\sigma_X}{d_{cor}\cdot\sigma_{dh}}$ for the three tested void filling techniques.

| Approach | Mean of $X$ (m) | *p*-Value | $\sigma_X$ (m) | $\frac{\sigma_X}{d_{cor}\cdot\sigma_{dh}}$ |
|---|---|---|---|---|
| SL-5 | −0.799 | 0.211 | 9.285 | 1.17 |
| CL-BL | −0.944 | 0.140 | 9.289 | 1.11 |
| CL-LO | 20.760 | 0.371 | 125.142 | 1.02 |

For all tested methods, the *p*-value is greater than 0.05. Thus, we cannot reject the null hypothesis, as expected. The standard deviation of $X$ is also provided in Table 4 and we can check that it approximately satisfies

$$\frac{\sigma_X}{d_{cor}\cdot\sigma_{dh}} \approx 1 \tag{17}$$

as expected. The 95% confidence interval from Equation (16) is depicted by inclined gray dashed lines in Figure 7 for $N > d_{cor}^2$. For glacier areas with $N$ in the order or less than $d_{cor}^2$, Equation (17) does not

need to apply. Still, in this case, we can approximately upper bound the standard deviation of $\overline{dh_{gl}}$ by $\sigma_{dh}$ so that for small $N$, we can hope that $\overline{dh_{gl}}$ might approximately satisfy

$$\overline{dh_{gl}} = \overline{dh} \pm 2\sigma_{dh} \text{ for } N \leq d_{cor}^2. \tag{18}$$

The bound (18) is depicted by the horizontal gray dashed lines in Figure 7. Empirically, we can see that indeed, for nearly all glaciers, $\overline{dh_{gl}}$ is within the bounds provided by Equations (16) and (18). We stress that while for $N \gg d_{cor}^2$, Equation (16) provides statistically sound confidence intervals, and for small $N$, our bound is just a rule of thumb because the central limit theorem on which Equation (16) is based does not apply. Indeed, empirically, we see that for large $N$, the bounds seem tight, as expected from the central limit theorem, and for small $N$, they are very conservative (but empirically valid).

To validate our observations numerically, we computed the fraction of $\overline{dh_{gl}}$ within the proposed bounds (the gray dashed lines in Figure 7). We performed a computation for $N > 10d_{cor}^2$ and a separate computation for all $N$. For all three methods, this fraction, denoted as $f^{int}$ in Figure 7, is slightly below 95% (for $N > 10d_{cor}^2$ and for all $N$). Considering that $\frac{\sigma_X}{d_{cor} \cdot \sigma_{dh}}$ is slightly larger than 1 (Table 4), we conclude that $d_{cor}$ is slightly underestimated, explaining why the values of $f^{int}$ are slightly below 95% for $N > 10d_{cor}^2$.

Finally, combining (17) and (18), the average elevation change offset in the void-filled regions $\delta_{dh}^V$ can be conservatively upper bounded by

$$\left| \delta_{dh}^V \right| \leq \left| \overline{dh} \right| + 2d_{cor} \frac{\sigma_{dh}}{max\left( \sqrt{N}, d_{corr} \right)}. \tag{19}$$

Our study region covers a wide sample of different glacier types and elevation change patterns. Thus, we claim that the findings can be transferred to other glaciated regions, assuming a comparable performance of the void filling approaches and void distribution. In order to test this claim, we used the shearlet void filling results and split the study region in two sections with non-overlapping sets of glaciers in each section. We computed $\overline{dh}$ and $\sigma_{dh}$ for the first section. Both values are only slightly different to those obtained for the whole study region. Then, we used the $\overline{dh_{gl}}$ values of the second section to carry out the same analysis as for the whole study region, but using $\overline{dh}$ and $\sigma_{dh}$ from the first section (Figure S5). We found that 95% of the $\overline{dh_{gl}}$ values are within the bounds given by Equation (19). The results strongly support our claim that the proposed estimation of the maximum average elevation change offset in the void-filled regions is transferable.

A common, but unproven, assumption to account for the uncertainty due to void filling in elevation change fields is to apply a multiple ($f_V$ of 2 to 5) of the error of the measured elevation changes $\delta_{dh}$, typically estimated on ice-free areas, to the void areas [2,4,5,7,20,29,30]. The subsequent volume change uncertainty on the void-filled region is computed as

$$\delta_{dV}^V = f_V(1 - p) \cdot A \cdot \delta_{dh} \tag{20}$$

where $A$ is the total glacier area and $p$ is the percentage of the glacier area covered by measurements.

The analysis presented in this section allows us to estimate the uncertainty caused by the void filling in a principled and more accurate way. Specifically, we propose an adjusted version of Equation (20) to estimate the uncertainty of the volume change on the void-filled areas:

$$\delta_{dV}^V = (1 - p) \cdot A \cdot \left( \delta_{dh} + \delta_{dh}^V \right). \tag{21}$$

To compare the two approaches at estimating the impact of void filling on glacier volume change computations, we selected the elevation change information provided by [6] for the glaciers in the northern wet outer tropics of Peru (called Subegion R1 in [6]) in the period 2013–2016. The authors reported a measured elevation change uncertainty $\delta_{dh}$ of 0.846 m (0.228 m/a), $A$ of 774.9 km$^2$ and a

coverage $p$ of 80% (number of void pixels $N = 172,200$). Applying Equation (20) with $f_V = 2$ leads to an $\delta_{dV}^V$ of 0.262 km$^3$. Using Equation (21) with $\delta_{dh}^V$ obtained from Equation (19) for $\delta_{dV}^V$, we obtain 0.138, 0.138 and 0.229 km$^3$ for shearlet (nscales = 5), bilinear and local hypsometric void filling methods, respectively. The contribution by $\delta_{dh}$ to $\delta_{dV}^V$ (0.131 km$^3$) dominates. Thus, the additional uncertainty caused by the void filling is negligible for the shearlet and bilinear approaches. For the local hypsometric approach, a contribution of ~37% to $\delta_{dV}^V$ by the void filling is found. All $\delta_{dV}^V$ estimates based on Equation (21) are smaller (53–87%) than the estimation based on Equation (20), leading to overall lower uncertainties in glacier mass balance computations.

Based on our proposed relation (Equation (16)), Equations (19) and (21) provide estimates for the average elevation change and volume change offset at the void-filled areas based on an empirical analysis using the estimates for $\overline{dh}$, $\sigma_{dh}$ and $d_{cor}$ obtained in this study for the tested void filling approaches.

## 5. Conclusions

The number of available DEMs is steadily increasing, facilitating the computations of geodetic glacier mass balances on extended spatial and temporal scales. Nearly all DEM data sets have voids or contain areas with unreliable data that should not be considered for further analyses. In order to minimize biases in the volume and mass change estimations, the data voids in the derived elevation change maps need to be filled in. Further, data voids frequently limit the direct assimilation of remote sensing products into ice dynamic models. Several methods were developed using spatial statistics or interpolation techniques to fill voids. In this study, we applied three novel void filling methods (Telea, Navier–Stokes and shearlet void filling) on artificially voided glacier elevation change data in southwest Alaska and compared these methods with results from classical techniques (bilinear, local and global hypsometric void filling). Moreover, we developed a new approach for estimating the uncertainties of glacier volume change measurements due to void filling.

The performance of the void filling methods was evaluated by comparing their outputs with the original void-free data set. The hypsometric approaches, in particular the global hypsometric approach, showed the worst overall performance. The filling of the voids with discrete values in the respective elevation intervals causes strong local offsets, and may generate relatively large overall offsets, as compared to the other methods. Only for one specific, difficult void setting (Center setup, Terminus void), the hypsometric approach was the best, which happened because the other methods did not have enough data to be well constrained. Many studies relied on the hypsometric approaches to fill voids in elevation change maps. However, based on our analysis, we cannot recommend using it, but if one uses it, the local hypsometric method should be preferred.

The shearlet method had some difficulties when applied on small study sites with relatively large voids; however, in the large-scale analysis, it showed a very good performance. Consequently, we only recommend to use this technique when sufficient coverage of the area around each void (or, better yet, some scattered data inside the void) is available to constrain the method; further base functions with a sufficiently large extent (as compared to the void sizes) should be contained in the shearlet dictionary.

The bilinear void filling generated very good results, comparable to shearlet void filling, in the large-scale analysis, but also had some difficulties accurately reconstructing the data for large voids. This approach needs sufficient data around the void for a good performance, and thus can lead to bad results if, e.g., a whole glacier section is missing (Center setup, Terminus void).

Both the Navier–Stokes and Telea approaches show reasonable performance on all test setups. Navier–Stokes void filling generates slightly, but not statistically significantly, better results than the Telea technique. The size of the search radius to constrain the approaches has no statistically significant impact on the quality of the results for the Navier–Stokes approach, but for the Telea approach, a statistically significantly higher variance exists for larger search radii.

The most stable overall performance was revealed for Navier–Stokes void filling considering all test setups. However, if there are sufficient data to constrain the shearlet and bilinear approaches,

as for the regional analysis (Correlation setup) in this study, they lead to the best results. Even though the shearlet approach is slightly better than the bilinear method, we recommend the bilinear method to be used in glacier mass balance computations (if it is well constrained, see above). It is by far much less computationally expensive than the shearlet approach, or the Navier–Stokes and Telea methods, and can be easily implemented in existing processing routines.

It is not straightforward to estimate the uncertainty of glacier volume change computations induced by void filling because reference data are not available to analyze the offsets generated by the void filling. Often a rule of thumb estimate is used to account for potential void filling offsets, which is not grounded in statistical considerations. In this study, we proposed and validated a precise relation to estimate the uncertainty of the void filling and the impact on glacier volume change estimations. The formula is based on the number of void pixels in the area of interest and method-dependent parameters carefully estimated in this paper. Since our study region consists of a heterogeneous sample of different glacier types, sizes and elevation change patterns, our relation to estimate the void filling uncertainties can be transferred to other study sites, a claim that we tested empirically. A comparison with the commonly used previous approach to uncertainty estimation revealed a strong reduction in the uncertainty on the void areas. Hence, using our method will lead to lower uncertainties in glacier mass balance computations.

Besides optical or SAR imagery, air- or spaceborne laser altimetry is another important technique to measure glacier surface elevation changes, in particular since the launch of ICESat-2 in 2018. However, only point measurements can be obtained by laser altimetry. Depending on the sensor, study region and weather conditions, such point measurements can be very sparsely and unequally distributed on the glacier area, making it more difficult to extrapolate the information throughout the glacier area. Thus, a detailed testing and analysis of different techniques, similar to this study, would be desirable to evaluate the accuracy of extrapolation techniques and to provide method recommendations. Additionally, maps of glacier surface velocities have also become a standard remote sensing product. These maps are used to analyze glacier dynamics or are assimilated in glacier models. Data voids in the velocity fields also limit their applicability. Inspired by this study, an interesting future project is a detailed evaluation and analysis of void filling approaches for glacier velocity data sets, in order to identify limitations and to find the most suitable methods to accurately fill that type of void.

Finally, we also would like to stress that, for the sake of reproducibility and transparency, for void-filled products, one should always provide metadata indicating the void-filled areas.

**Supplementary Materials:** The following are available online at http://www.mdpi.com/2072-4292/12/23/3917/s1, Figure S1: Void filling results of Center setup, Figure S2: Void filling results of Juneau setup, Figure S3: Hypsometric distribution of glacier area and surface elevation changes of Center setup, Figure S4: Hypsometric distribution of glacier area and surface elevation changes of Juneau setup, Figure S5: Scatter plot of $\left( \overline{dh_{gl}} - \overline{dh} \right)$ versus $\sigma_{dh} / \sqrt{N}$ for the Shearlet (nscales = 5) approach.

**Author Contributions:** T.S. designed and led the study, processed the data, performed statistical analysis and wrote the manuscript. V.I.M. proposed the shearlet inpainting method, performed statistical analyses and proposed the formula for accuracy estimation at different scales. F.H. implemented the shearlet inpainting and processed the shearlet results. E.B. and M.H.B. initiated and supervised the project. All authors supported the writing of the manuscript. All authors have read and agree to the published version of the manuscript.

**Funding:** This work was financially supported by the FAU Emerging Fields Initiative grant TAPE and the STAEDLER Foundation.

**Acknowledgments:** The digital elevation models, elevation change maps and adjusted glacier outlines were kindly provided by Robert McNabb.

**Conflicts of Interest:** The authors declare no conflict of interest. The funding sponsors had no role in the design of the study; in the collection, analysis, or interpretation of data; in the writing of the manuscript, and in the decision to publish the results.

**Data and Code Availability:** The input data and void filling results are available at PANGAEA (DOI registration in progress). The code to generate void-filled elevation change data sets and to reproduce the statistical analysis is available at GitHub (https://github.com/tseehaus/inpainting-dhdt).

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
