# Peer review of "Novel Techniques for Void Filling in Glacier Elevation Change Data Sets"

_remotesensing, doi:10.3390/rs12233917_

Round 1

Reviewer 1 Report

In their Remote Sensing manuscript “Novel techniques for void filling for glacier elevation change data sets” Seehaus et al. evaluated three classical and three novel approaches for interpolating voids in elevation change maps. They further developed a formula to estimate uncertainties from void filling which can be employed for other study sites and is hence useful for others.

First of all I want to thank the authors for this nice study where one can tell that a lot of effort has gone into it. Unfortunately the latter is not only valid for its content but also for its length which sometimes makes it hard to follow. I have the feeling the study needs to be condensed somehow to keep the reader engaged. This could potentially be done by moving some of the content into the Supplement. In the following I have some questions and suggestions followed by a couple of specific comments.

General remarks:

1. Is there a reason why the authors did not test kriging as another classical interpolation method? See for example Pieczonka & Bolch 2015.

2. This might be a little bit beyond the scope of the study, but for an ICESat like setup hypsometric approaches might still be the only valid interpolation method. This will certainly be of interest for new ICESat-2 based studies. I could think of an additional setup mimicing ICESat-2 tracks over the glacierized area, resulting in much more data voids than valid measurements. If not included in the analysis this point should be addressed in the discussion and conclusions.

Specific comments:

Line 24: as far as I understand all interpolation methods are applied to elevation change maps and not to glacier elevation data sets. Please be clear on this point. In line with this, is there a reason why the interpolation was done solely on elevation change maps and not on the original DEMs prior to differencing?

Line 46: 43% and 97% are values from the literature. I therefore think “can vary” is a little bit misleading. However, it might be interesting to actually find a lower threshold of voids (see also my general remark 2 on this).

Lines 55-57: I suggest to focus on elevation changes not on velocity products. This could potentially be removed to shorten the introduction.

Line 56: lead.

Lines 62-91: large parts of this section are also described in the Method section, which give the potential to further shorten the introduction.

Line 75: grouped.

Line 122: maybe the authors can clarify that the 2.5% voided area in the SRTM data can not be used for validation and therefore it is not included in their analysis. In a real mass balance scenario these voids would also be interpolated.

Lines 281-342: could potentially be moved to the Supplement.

Lines 346: the default setting of r.fillnulls uses regularized spline interpolation.

Line 409: equation 14, I think.

Lines 424-425: which is the best approach is not yet clear.

Lines 696-697: see general remark 2.

Figure 2: I think a link to Figure S1 should be included in the caption.

Table 3 and Figure 5 show almost the same. I suggest to move Table 3 to the Supplement.

Figure 3: what is shown here mean elevation change and standard deviation? Please state in the caption.

Figure 4: is there a reason why the local hypsometric approach is not listed here and also not shown in Figures S1 and S2? Please explain. I also find this figure hard to read and suggest to rather show a scatter plot similar to Figure 5.

Table 4: do the authors really show values for the global hypsometric method? I am a bit confused as in line 598 the local hypsometric method is mentioned (which is also shown in Figure 6).

Lines 732-737: I probably would remove this from the Conclusions as it is beyond the scope of the current study.

Additional references:

Pieczonka, T. and Bolch, T.: Region-wide glacier mass budgets and area changes for the Central Tien Shan between ∼1975 and 1999 using Hexagon KH-9 imagery, Global Planet. Change, 128, 1–13, https://doi.org/10.1016/j.gloplacha.2014.11.014, 2015.

Author Response

See file attached

Reviewer 2 Report

This is a well-written and very thorough paper. It tackles a method (void filling)  that is used substantially in the glaciology community, and other disciplines. The authors very carefully analyze different DEM-filling methods. It's possible the details are only of-interest to a small group of researchers, but the conclusions of this study provide an important guide for many. My only comments are small typos (thank you authors for such a well-written paper, makes a reviewer's job easy)!

56: leads -> lead

75: group -> grouped

Figure 1: setup boxes are really hard to see - can you make them a bright color?

Figure 4: Font size is really small.

Author Response

see file attached

Reviewer 3 Report

Review of manuscript entitled Novel Techniques for Void Filling for Glacier Elevation Change Data Sets by T. Seehaus, V. I. Morgenshtern, F. Hübner, E. Bänsch, and M. H. Braun. The manuscript is submitted for publication in Remote Sensing.

Summary.

The objective of the described study is to test several methods for void filling of DEMs to be used in geodetic glacier mass estimates. The study investigated the performance of several common/classic approaches (bilinear interpolation, local and global hypsometric method) as well as several novel approaches (Telea, Navier-Stokes and Shearlet). Their study region is 6408 km2 of glacier area in southeast Alaska, which is covered by two void-free DEMs. Their findings support a recommendation of using bilinear void filling methods.

General comments:

The manuscript tackles an interesting, important and timely issue. As more and more remote sensing data become available, we need studies that confirm the performance of e.g. void filling techniques.

The manuscript is generally very well writte. The scope, objective and arguments are clearly communicated.

A critical point in the study setup is the generation of the artificial voids, and it is my opinion that this has been done well, taking into consideration that they should be similar to voids that are evident in data due to e.g. cloud cover or sensor saturation or data gaps between satellite acquisitions.

The manuscript includes some quite technical information about which libraries and programs have been used and what the computational was. I actually appreciate this information as it is crucial for other potential users to know.

Main issues/questions:

It seems to me that this study builds on the McNabb et al (2019) published paper. I think that the authors should be clearer on the connection between the previously published paper and the current one.

It would strengthen the manuscript could be a bit clearer in their argumentations for testing the three novel approaches. Why were these chosen above others? What is the reasoning for why you think that they were worth testing?

About the Telea approach: Out of curiosity how do you avoid dis-continuities at the center of the void when propagating from the edge of the void and fill in in towards the center? Is this never an issue?

You write “no-data” pixels outside the glaciers are treated as zeros.’ In L. 301-302 You write that this might affect the void filling, which is clear. But why not assign the surroundings no value (NaN) so that they are not used in the analysis, instead of extending the void filler onto a buffer zone?

Generally, the results and discussion section is very long and very dense on information. I am wondering if maybe some of the information can be moved to the suppl. Material. In my personal opinion e.g. Figures 6 and 7 could be moved to the suppl. Material while I would find it relevant to actually show some void filling result in the main text (they are now only found in the suppl. material)

Minor issues:

l 64-65 Can you elaborate on the pros and cons of void filling individual DEMs or the derived elevation change maps?

  1. 73-74. You state that ‘…so that the interpolation is not identifiable bya viewer’. But is it not an advantage that the end-user knows which regions/pixels are observations and which are estimated via void-filling? So, even if it appears seamless it should be identified somehow in the end-product.

L 85. Can you explain what is meant by ‘highly compressible’ in this context?

L 90. What methods do you specifically refer to here? There are many different neural network methods, I assume?

  1. 97. Can you be a bit more specific about what you mean with inpainting results? This is not clear to me at this point.

  1. 108. The area that you state here does not match the area you write in the abstract. Which one is correct?

  1. 145- I suggest that you also refer back to Fig. 1, where these are outlined for clarity.

Table 1. This table can to my opinion be omitted, and the naming be described in a few sentences in the main text.

  1. 368 Isn’t it rather an error in glacier volume change estimation than in glacier mass change?

Figure 2. The polygons are not shown here as closed polygons, which makes the figure confusing. Are the polygons larger than the figures?

Figure 3: which dataset is this based on? The entire area?

I think that it would improve the results section if a few of the figures from the Suppl. Material were included as examples in the main text.

Figure 4: It is difficult to read the text in this figure. Both a result of size and resolution.

Figure 5. I suggest that you colour code the figure, as it is difficult to see which result belong to which technique

Figure 6: I do not understand the legend of these plots. Please clarify. Same for Figure 7

  1. 693- 695 I do not agree with your conclusions that ‘Only for one specific, difficult void setting (Center setup, Terminus void), the hypsometric approach was the best, which most likely happened by chance plus other methods did not have enough data to be well constrained.’ It is not (in my opinion) by chance… this means that the hypsometric approach has its merit in some cases.

Technical corrections:

When you use references that you spell out, please indicate the year also. E.g. l. 62 McNabb et al. -> McNabb et al., (2019)

  1. 75: group -> grouped
  2. 83: approaches -> approach
  3. 147: mainly RGI60-01.20686 -> mainly the RGI60-01.20686
  4. 355 using 2-98% -> using a 2-98%
  5. 449: are -> is
  6. 588 guarantees -> conclusions ?

Author Response

see file attached
